# The Role of Complementary Feeding Practices in Addressing the Double Burden of Malnutrition among Children Aged 6–23 Months: Insight from the Vietnamese General Nutrition Survey 2020

**DOI:** 10.3390/nu16193240

**Published:** 2024-09-25

**Authors:** Pui Yee Tan, Somphos Vicheth Som, Son Duy Nguyen, Do Tranh Tran, Nga Thuy Tran, Van Khanh Tran, Louise Dye, J. Bernadette Moore, Samantha Caton, Hannah Ensaff, Xiaodong Lin, Geoffry Smith, Pauline Chan, Yun Yun Gong

**Affiliations:** 1School of Food Science and Nutrition, Faculty of Environment, University of Leeds, Leeds LS2 9JT, UK; p.y.tan@leeds.ac.uk (P.Y.T.); somphosvichethsom@gmail.com (S.V.S.); l.dye@sheffield.ac.uk (L.D.); j.b.moore@leeds.ac.uk (J.B.M.); h.ensaff@leeds.ac.uk (H.E.); 2Section of Infectious Diseases, Department of Health Sciences, Vrije Universiteit Amsterdam, 1081 HV Amsterdam, The Netherlands; 3Division of Human Nutrition and Health, Wageningen University and Research, 6700 HB Wageningen, The Netherlands; nguyenduysonvdd@gmail.com; 4Department of Nutrition Surveillance and Policy, National Institute of Nutrition, 48B Tang Ba Ho, Hai Ba Trung District, Hanoi 100000, Vietnam; thanhdo.tran@gmail.com; 5Department of Micronutrient, National Institute of Nutrition, 48B Tang Ba Ho, Hai Ba Trung District, Hanoi 100000, Vietnam; thuynga1997@gmail.com (N.T.T.); trankhanhvan.nin@gmail.com (V.K.T.); 6Institute for Sustainable Food and Department of Psychology, University of Sheffield, Sheffield S1 4DP, UK; 7Sheffield Centre for Health and Related Research (SCHARR), School of Medicine and Population Health, University of Sheffield, Sheffield S1 4DP, UK; s.caton@sheffield.ac.uk; 8Global Sustainable Development, University of Warwick, Coventry CV4 7AL, UK; xiaodong.lin@warwick.ac.uk; 9International Life Sciences Institute (ILSI) Southeast Asia Region, 9 Mohamed Sultan Road, #02-01, Singapore 238959, Singapore; geoffsmith@ilsisea.org.sg (G.S.); paulinechan@ilsisea.org.sg (P.C.)

**Keywords:** complementary feeding practices, minimum dietary diversity, minimum acceptable diet, dietary quality, malnutrition, micronutrient deficiencies, infants, Vietnamese

## Abstract

**Background/Objectives**: Optimal infant and young child feeding (IYCF) practices are crucial to addressing the double burden of malnutrition (DBM), encompassing undernutrition (including micronutrient deficiencies) and overnutrition. This study examined the demographic and socioeconomic determinants of IYCF practices, and their impacts on the DBM among 2039 Vietnamese children aged 6–23 months from the General Nutrition Survey 2020. **Methods:** Thirteen IYCF indicators recommended by the WHO/UNICEF were evaluated. Associations between IYCF indicators and outcome variables were assessed using logistic regressions. **Results:** The prevalence of stunting, underweight, and overweight subjects was 10.9%, 5.6%, and 3.1%, respectively. Low serum zinc affected 56.7% of children, while 14.3% had low serum retinol, 31.2% had anemia, and 34.6% had iron deficiency (ID). Only 36.7% of children achieved minimum dietary diversity (MDD), and 29.0% achieved the minimum acceptable diet (MAD). Children from the younger age group (6–11 months), ethnic minorities, those living in rural/mountainous regions, and poorer wealth quintiles had reduced odds of meeting IYCF criteria, including MDD and MAD. Infants meeting MDD had reduced odds of stunting [adjusted odds ratio (95% confidence intervals): 0.61 (0.41, 0.92)], and ID [0.69 (0.54, 0.88)]. Children meeting MAD had reduced odds of anemia [0.72 (0.57, 0.91)], ID [0.66 (0.52, 0.84)], and low serum retinol [0.63 (0.41, 0.99)]. Continued breastfeeding (12–23 months) reduced the odds of being underweight [0.50 (0.27, 0.92)] and of having low serum zinc [0.70 (0.52, 0.96)]. Adequate minimum milk feeding frequency had increased odds of being overweight [3.33 (1.01, 11.09)]. **Conclusions:** Suboptimal IYCF practices were significant predictors of the DBM among Vietnamese children, with evident age-specific, geographical, and socioeconomic disparities.

## 1. Introduction

Many low- and middle-income countries (LMICs), including Vietnam, are now facing a double burden of malnutrition (DBM), with the coexistence of overnutrition, undernutrition, and micronutrient deficiencies (MNDs) [1]. Despite notable efforts and progress in reducing malnutrition among children aged under 5 over the past decades, malnutrition remains a major public health challenge in Vietnam. In Vietnam, the prevalence of stunting among children aged 6–23 months increased from 9.0% in 2010 [2] to 17.5% in 2015 [3]. The Southeast Asia Nutrition Survey (SEANUTs) in 2011 also reported that 5.3% and 13.0% of children aged 6–23 months in rural areas were stunted and underweight, respectively, whereas 4.7% and 2.7% of children in urban areas were overweight or obese, respectively [2]. Half of the children in Vietnamese rural areas were affected by anemia (54.3%), double the prevalence found in urban areas (25.9%) [2].

Evidence suggests that suboptimal infant and young child feeding (IYCF) practices are the key determinants of child malnutrition in LMICs, particularly in the poor, remote, rural areas where there is limited access to healthcare services and shortfalls in human, financial, and information resources [4]. Adequate nutrition during infancy and early childhood, especially the first 1000 days of life from conception to 2 years of life, is essential to ensure optimal growth and development of children [5]. Early life nutrition could outweigh the genetic predisposition on linear growth patterns and lead to irreversible impairment in children’s growth and development [6]. Moreover, poor infant nutrition has been significantly associated with increased risk of children being overweight and obese, and with an increased risk of non-communicable diseases (NCDs) in later life [7]. Thus, promoting optimal feeding practices during infancy is a crucial strategy to protect against both undernutrition and overnutrition, and to alleviate chronic diseases in the short and long term.

The World Health Organization (WHO) and UNICEF have developed 17 global standard indicators for the use in population-based nutrition surveys to assess IYCF practices across countries [8]. Suboptimal IYCF practices are often observed in countries with the highest burden of malnutrition, especially in LMICs. Only 21%, 56%, and 10% of the 80 LMICs had a prevalence > 50% for minimum dietary diversity (MDD), minimum meal frequency (MMF), and minimum acceptable diet (MAD), respectively. The WHO/UNICEF recommended IYCF indicators were first officially introduced into Vietnam’s national nutrition surveillance in 2010 with support from Alive & Thrive (A&T) and UNICEF [8,9]. In 2011, the A&T Project reported that only 44–73% and 33–52% of different ethnic minorities received MDD and MAD compared to the major Kinh ethnic children (86% and 75%), respectively, suggesting demographic variation in feeding practices across Vietnam [10].

IYCF practices are crucial for optimal child growth, yet limited data are available in Vietnam. Understanding IYCF practices and their demographic and socioeconomic determinants is essential to informing targeted intervention to address all forms of child malnutrition. Therefore, in this study, we aimed to utilize nationally representative data from the recent Vietnamese General Nutrition Survey (GNS) 2020 to evaluate IYCF practices, including dietary diversity and dietary quality, and their associated demographic and socioeconomic determinants, as well as their impacts on DBM, including undernutrition, overnutrition, and MNDs (anemia, iron deficiency (ID), low serum retinol, and low serum zinc) among Vietnamese children aged 6–23 months. The findings from this study provide valuable and updated data for policy makers and researchers to inform targeted interventions to promote IYCF practices and nutritional outcomes, not only in Vietnam but also in other Southeast Asian and LMIC countries with similar contexts.

## 2. Materials and Methods

### 2.1. Study Design and Study Population

This study analyzed data collected from the nationally representative GNS 2020, which is conducted every 10 years. The survey methodology has been detailed in a previous publication [11]. Participants were recruited using a multi-stage cluster sampling design to minimize selection bias. Census enumeration areas (EAs) served as the primary sampling units (PSUs). EAs were selected from 143 communes across 25 provinces to represent 6 geographical areas, including (i) the northern midlands and mountain areas, (ii) the Red River Delta (including Hanoi Capital), (iii) the north central and central coastal area, (iv) the central highlands, (v) the southeast, and (vi) the Mekong River Delta (including Ho Chi Minh City). EAs were further stratified by area of residence, and subsequently by targeted population and sex. Eligible individuals from each target population group were randomly chosen from the selected EAs. A total sample of 20,864 individuals were enrolled in GNS 2020. For this study, individuals were excluded if they had missing demographic data, or if they were aged <6 months, or >23 months. The final analysis included 2039 children aged from 6 to 23 months who met the inclusion criteria and had complete demographic and socioeconomic data (Appendix A; participant flow chart).

### 2.2. Anthropometric Measurement

The body weight and height/length of the children were measured. According to the WHO Child Growth Standard, weight-for-age z-scores (WAZ), height-for-age z-scores (HAZ), and weight-for-height z-scores (WHZ) < −2 standard deviation (SD) were used to define underweight status, stunting, and wasting, respectively [12]. The overweight status was defined as WHZ > +2 SD, and ≤+3 SD, and obesity was defined as WHZ > +3 SD. Children with HAZ and WHZ < −6 SD, or >+6 SD, and/or children with WAZ < −6 SD, and >+5 SD, were flagged as outliers and were removed [12].

### 2.3. Biochemical Assessment and Definitions of Micronutrient Deficiencies

Blood samples were collected as previously described [11]. The blood micronutrients investigated in this study were selected based on three main criteria: (1) micronutrient biomarkers that were measured and available in the GNS 2020 dataset, (2) high prevalence and severity of deficiencies for these micronutrients in Vietnam, and (3) alignment with the priorities of Vietnamese government based on the nutritional indicators and targets outlined in the National Nutrition Strategy 2025 and 2030 [13]. As a result, four micronutrient deficiencies, including anemia, iron deficiency, vitamin A deficiency (now referred as “low serum retinol [14]), and zinc deficiency (now referred as “low serum zinc” [15]), were selected.

Hemoglobin (Hb) was measured immediately after blood sample collection using a HemoCue^®^ Hb 301 analyzer (HemoCue, Angholm, Sweden). Serum ferritin, transferrin receptor (sTfR), C-reactive protein (CRP), α-1-acid glycoprotein (AGP), and retinol binding protein (RBP) concentrations were measured by VitMin Lab ELISA assays (VitMin Lab, Willstaett, Germany). Accuracy was confirmed using controls from the Centers for Disease Control and Prevention, USA, and the National Institute of Biological Standards and Control, UK. Serum retinol concentrations were determined by reverse-phase liquid chromatography with tandem mass spectrometry (3000 Qtrap, Sciex, Framingham, MA, USA). Serum zinc concentration was analyzed by a flame atomic absorption spectrophotometer (GBC, Avanta+, Keysborough, Australia) using trace element-free procedures and powder free gloves, and the results were verified using reference materials (Liquicheck, Bio-Rad Laboratories, Hercules, CA, USA). To avoid measurement bias, within-assay and between-assay variability tests were performed for quality control of the biomarker measurements. The within-assay CVs for serum ferritin, zinc, retinol, CRP, and AGP ranged from 2.9 to 7.1%, and between-assay variability was <10% for all the biomarkers.

Anemia, ID, iron deficiency anemia (IDA), and low serum retinol were defined according to the WHO guidelines. Anemia was defined as Hb concentrations < 110 g/dL [16]. Serum ferritin concentrations < 12 µg/L and <30 µg/L were used to define ID among non-inflamed children and inflamed children, respectively. Inflammation was defined as CRP > 5 mg/L and/or AGP > 1 g/L [17]. The coexistence of anemia and ID was defined as IDA. Serum retinol concentration < 0.70 µmol/L was used to define low serum retinol [14]. Low serum zinc was defined as a serum zinc concentration < 9.9 μmol/L (morning, non-fasting) or <8.7 μmol/L (afternoon, non-fasting) according to the International Zinc Nutrition Consultative Group (IZiNCG) [15]. The total number of MNDs included anemia, ID, low serum retinol, and low serum zinc.

### 2.4. Breastfeeding and Complementary Feeding Practices

Mothers were asked to recall the types of food and drink and number of times that their children consumed in the past 24 h prior to the survey using a food frequency questionnaire [8]. Of the 17 IYCF indicators recommended by WHO/UNICEF, we selected 13 indicators that assess IYCF practices for infants aged 6–23 months [8]. The excluded four indicators (i.e., exclusive breastfeeding under six months, mixed milk feeding under six months, exclusively breastfed for the first two days after birth, and infant feeding area graph) pertain to breastfeeding practices for infants aged 0–5 months and are not relevant to our study population; therefore, they were omitted. Definitions for each of the 13 selected IYCF indicators are provided in Table 1.

### 2.5. Other Demographic and Socioeconomic Determinants

A structured questionnaire was used to collect demographic and socioeconomic data from the parents/caregivers, including age, sex, ethnicity, geographical area, and area of residence. The household wealth index score was calculated using data on household’s ownership of selected assets, materials used for housing construction, types of water access, and sanitation facilities collected in our survey, using principal component analysis (PCA) following the Demographic and Health Survey (DHS) guidelines [18]. Each household was then assigned a wealth index score based on the PCA factor loadings and was then further grouped into the following quintiles for analysis: poorest (Q1), poorer (Q2), middle (Q3), richer (Q4), and richest (Q5). This household wealth index was used as a proxy indicator for children’s socioeconomic status.

### 2.6. Statistical Analysis

All statistical analyses were conducted using STATA software (version 18; Stata Corp LLC, Cary, NC, USA), All analyses were performed based on sampling weights, using svyset commands to adjust for individual sampling weight, clustering, and stratification. As most of the missing data occurred in outcomes (<15%) and only <0.5% missing data occurred in predictors, complete-case analysis was preferred in this case, and data imputation was not performed [19]. Continuous variables and categorical variables were reported as the mean ± SD and number (percentage), respectively. Differences in demographic, socioeconomic, and anthropometric parameters, as well as biomarkers of micronutrients between sex, were assessed using Chi-square test of association and the *t*-test, for categorical and continuous data, respectively. Bivariate logistic regressions were performed to determine the associations between IYCF practices and demographic and socioeconomic determinants. Multivariate logistic regressions were performed to assess the associations between IYCF practices and different forms of malnutrition. Age, sex, area of residence, wealth index, and inflammation showed significant associations with IYCF practice indicators, and were further adjusted in the multivariate logistic regression analyses. Ethnicity and geographical area were not included in the models for adjustment due to multicollinearity. The crude odds ratio (OR) or adjusted OR (AOR) with its respective 95% confidence intervals (CI) were reported. A1 *p* value < 0.05 was considered statistically significant. Sensitivity analyses were performed (i) by comparing the randomly selected subsamples (50% of the total population) within the total samples, and (ii) testing two logistic regression models with and without adjustments.

## 3. Results

### 3.1. General Characteristics

The general characteristics of the study participants (*n* = 2039) by sex are reported in Table 2. The average age of the Vietnamese children was 14 ± 5 months. The prevalence of stunting, underweight, and wasting children was 10.9%, 5.6%, and 3.4%, respectively, whereas a smaller percentage of the children were overweight or obese (2.5% and 0.6%, respectively). The number of observations available for each serum micronutrient biomarker were as follows: Hb *n* = 1822, ferritin and sTfR *n* = 1728; zinc *n* = 1820, and retinol *n* = 1784. The numbers and proportions of missing data are reported in Appendix A. Low serum zinc was found in more than half of the children (56.7%), and low serum retinol as found in about one-seventh of the children (14.3%). The prevalence of anemia, ID, and IDA was 31.2%, 34.6%, and 17.8%, respectively. High inflammatory status was found in 18.5% of the children. There were no significant differences in age, weight, height, area of residence, geographical area, ethnicity, and wealth quintiles between males and females (*p* > 0.05). Males had a significantly higher prevalence of stunting (14% vs. 7.7%; *p* < 0.01), being underweight (7.2% vs. 3.9%; *p* < 0.01), ID (40.0% vs. 28.8%; *p* = 0.01), IDA (21.0% vs. 14.3%; *p* < 0.01), and were more likely to have ≥2 MNDs (54.0% vs. 48.0%; *p* = 0.02) compared to females.

### 3.2. Adherence to IYCF Practices

Figure 1 illustrates the adherence to IYCF practice indicators among the study participants. The majority of the children were EvBF, 42.5% had EIBF, 60.7% were CBF at 12–23 months, and 80.4% had ISSSF at 6–8 months. Only 36.7% met MAD, 48.0% achieved MDD, and 29.0% achieved MMFF. About 66.9% of children had EFF and 38% had ZVF in the past 24 h. One-fifth of the children consumed sweet beverages (SWB) (20.7%), and 12.6% had engaged in UFC (sweet, fried, and salty foods).

### 3.3. Consumption of MDD’s Eight Food Groups

The most consumed food groups among the study participants were grains, roots, and tubers (95.4%), followed by flesh foods (72.1%), and breast milk (66.7%) (Figure 2). Over half of children consumed vitamin A-rich fruits and vegetables (62.6%), and dairy products (54.7%) in the past 24 h. In contrast, legumes, nuts, and seeds (6.8%), eggs (15.5%), and other fruits and vegetables (23.6%) were less frequently consumed.

### 3.4. Associations of Feeding Practices with Demographic and Socioeconomic Determinants

Compared to the younger infants (6–11 months), older infants had increased odds of meeting MDD [OR (95% CI): 1.19 (1.49, 2.46),] MMFF [4.26 (3.00, 6.06)], MAD [2.15 (1.61, 2.86)], EFF [1.69 (1.26, 2.25)], SBW [2.52 (1.60, 3.95)], and UFC [3.37 (2.53, 4.49)] (Table 3). Children living in rural and mountainous areas had decreased odds of meeting MDD [0.65 (0.49, 0.86); 0.44 (0.26, 0.74)], and MAD [0.77 (0.60, 0.98); 0.48 (0.23, 1.00)], respectively, compared to urban areas. Children living in mountainous areas were less likely to continue breastfeeding at 12–23 months [0.40 (0.22, 0.71)] and less likely to achieve EFF [0.52 (0.32, 0.85)]. Ethnic minorities had decreased odds of meeting MDD [0.65 (0.54, 0.79)], MMFF [0.56 (0.34, 0.91)], MAD [0.36 (0.20, 0.64)], CBF at 12–23 months [0.39 (0.21, 0.71)], and EFF [0.41 (0.29, 0.57)], and also showed increased odds of having ZVF [1.86 (1.32, 2.63)].

Compared to the poorest quintile, children from the richest quintile had increased odds of achieving MDD [2.69 (1.28, 5.67)], MMFF [2.23 (1.41, 3.54)], and CBF at 12–23 months [2.32 (1.04, 5.19)]. Moreover, children from the middle, richer, and richest quintiles had increased odds of achieving EFF (ORs ranging from 2.76 to 2.89), and reduced odds of meeting ZVF (ORs ranging from 0.42 to 0.56). Compared to children living in the northern mountains, those living in the Red River Delta and southeast had increased odds of achieving MDD [1.81 (1.06, 3.10); 2.39 (1.36, 4.21)] and CBF at 12–23 months [2.54 (1.14, 5.65); 2.63 (1.42, 4.89)]. When we analyzed each of the eight food groups, we identified significant demographic variation and socioeconomic disparities in food group consumption (Appendix A). Children from ethnic minorities, lower wealth quintiles, and those living in rural and mountainous areas were less likely to consume breast milk, dairy products, vitamin A-rich fruits and vegetables, and flesh foods.

### 3.5. Associations of Feeding Practices with Malnutrition and MNDs

In the adjusted regression models, children who achieved MDD [AOR (95% CI): 0.61 (0.41, 0.92)] and EFF [0.68 (0.48, 0.97)] showed reduced odds of stunting compared to those who did not achieve it, whereas children with ZVF had increased odds of stunting [1.55 (1.18, 2.04)] (Table 4). A marginal significant protective effect of SWB against the odds of stunting [0.64 (0.42, 0.99)] was observed. CBF at 12–23 months was associated with reduced odds of underweight [0.50 (0.27, 0.92)], and ISSSF at 6–8 months was associated with reduced odds of wasting [0.12 (0.04, 0.39)]. Non-breastfed children who achieved MMFF were associated with increased odds of being overweight [3.33 (1.01, 11.09). With respect to MNDs, children who achieved MDD had reduced odds of ID [0.69 (0.54, 0.88)], and IDA [0.63 (0.46, 0.88)] (Table 5). Similarly, children who achieved MAD showed reduced odds of ID [0.66 (0.52, 0.84)], IDA [0.56 (0.42, 0.75)], anemia [0.72 (0.57, 0.91)], and low serum retinol [0.63 (0.41, 0.99)]. Moreover, children who achieved MMF had reduced odds of ID [0.56 (0.38, 0.82)]. CBF at 12–23 months was associated with reduced odds of IDA [0.63 (0.42, 0.96)], and low serum zinc [0.70 (0.52, 0.96)]. ISSSF at 6–8 months was associated with reduced odds of low serum retinol [0.37 (0.16, 0.89)].

## 4. Discussion

Child malnutrition remains a major public health issue in Vietnam. Overall, 11%, 6%, and 3% of the infants aged 6–23 months were stunted, underweight and wasted, respectively. Low serum zinc was the most common MNDs observed (57%), followed by ID (34.6%), anemia (31.2%), and low serum retinol (14.3%), whereas overweight (2.5%) and obese (0.6%) children were of less concern in this age group. Suboptimal IYCF practices were observed in Vietnam. Only one-third of the infants aged 6–23 months had adequate MDD (36.7%), one-fourth had adequate MAD (29.0%), and half of the non-breastfed infants had adequate MMFF (56.0%). Several barriers to achieving optimal feeding practices in Vietnam have been previously identified; these include the lack of enforcement of marketing regulations on breast milk substitutes, insufficient counselling service on IYCF, inadequate knowledge among healthcare providers, cultural beliefs, and poor IYCF knowledge among mothers [20].

Compared to the Vietnam Alive & Thrive (A&T) Project 2011, our findings in 2020 suggest a lower prevalence of MDD (22.1% vs. 36.7% in ethnic minorities; and 39.5% vs. 86% in major Kinh ethnic children) [10]. It is noteworthy that our data were collected about 10 years after the A&T project. Furthermore, the MDD and MAD reported in the A&T Project were defined based on ≥4 out of 7 food groups without including breast milk (MDD-7FG), which may have led to an overestimation of MDD and MAD [21]. Vietnam is currently undergoing a dietary transition characterized by a shift from traditional diets towards a diet with increased consumption of meat, milk, sweetened beverages, and processed foods, accompanied by a decrease in the consumption of fruits and vegetables [22]. These trends align with our findings, as apart from staple foods, flesh foods primarily derived from meat, poultry, and organ meat (59.2% vs. 35.5% from fish only), vitamin A-rich fruits and vegetables, and dairy products were more frequently consumed by the Vietnamese children surveyed. Therefore, nutrition education initiatives and cooking sessions incorporating local recipes to encourage mothers and caregivers to include diverse foods, especially legumes, nuts and seeds, eggs, and other fruits and vegetables which were less commonly consumed in their children’s diets are highly encouraged.

Older infants were more likely to achieve MDD and MAD compared to younger infants, as the variety of food groups consumed increases as the infants age [23,24]. Additionally, previous studies reported that delayed introduction of complementary feeding is often associated with lower adherence to MDD and MAD among younger infants [24]. Other possible explanations include parents’ reticence to provide certain foods to younger infants, which itself may be linked to preconceptions that these foods, such as meat, green vegetables, pumpkin, and carrots, are inappropriate or difficult for them to digest, thus reducing their consumption among younger infants [25]. In line with previous findings from 80 LMICs, our findings demonstrated that dietary diversity and meal frequency were positively associated with household wealth status [4]. Wealthier families have stronger purchasing power and/or better access to variety of foods, making them less likely to experience food insecurity, poor dietary diversity, and infrequent meal feeding [26,27].

Our findings indicated that children from ethnic minorities, poorer families, and those living in rural/mountainous areas were less likely to access or consume nutrient-dense foods, such as meats, dairy products, eggs, and vitamin A-rich fruits and vegetables, resulting in poor dietary diversity and dietary quality. This may be due to economic constraints and sociocultural factors. Previous research reported that the price of meat and legumes has substantially increased between 1996 and 2015 in Vietnam, thus reducing their consumption by poorer households [22]. Other studies also reported that poor dietary quality and feeding practices among children living in rural and mountainous areas were often associated with household poverty [28], low maternal educational status, food insecurity, and fewer antenatal care visits and postnatal checkups [29].

Previous research showed that Vietnamese mothers’ feeding practices were deeply influenced by sociocultural factors and varied significantly across ethnic groups and regions [30,31,32]. Traditional beliefs can impact Vietnamese mothers’ breastfeeding practices, such as the fact that some Vietnamese mothers have traditional beliefs and practices about breastfeeding, such as the perception of colostrum as “dirty” and lacking nutritional value, leading to lower rates of EIBF [33]. Similarly, some food items, such as green leafy vegetables, are often considered “cold” and, thus, are not given to infants [24,30]. Families, especially the senior members (e.g., mothers, mothers-in-law, and grandmothers), friends, and neighbors play a significantly role in shaping mothers’ IYCF beliefs and practices [33,34]. Physical and emotional support from husband and families, as well as higher maternal education, have shown to increase mothers’ knowledge, belief, and confidence in adhering to optimal IYCF practices [34,35,36]. Additionally, the marketing of infant formula milk can adversely affect breastfeeding practices [37,38]. Therefore, it is important to incorporate sociocultural considerations into future research and interventions aimed at improving IYCF practices and child nutritional outcomes, particularly in resource-constrained and culturally diverse settings in Vietnam.

Our data show that suboptimal feeding practices and the poor dietary quality of complementary foods were the significant contributors to DBM in Vietnamese children. Infants who had ISSSF at 6–8 months showed reduced odds of wasting and low serum retinol. As breast milk alone cannot meet the needs of children aged over 6 months, timely introduction of complementary feeding is essential to provide sufficient energy and macro- and micronutrient intake from dietary sources as a child grows. Furthermore, infants who achieved CBF at 12–23 months showed reduced odds of being underweight, IDA, and having low serum zinc. This underscores the importance of continued breastfeeding in this context to provide additional energy and nutrients necessary to fulfil the dietary requirements of older infants, particularly in cases where complementary foods were scarce, or the diet lacked adequate nutritional quality. Although opposite results were found in a pooled analysis of 14 low-income countries, the authors postulated that this may be possibly due to the over-reliance of mothers on breastfeeding, which competed with dietary diversity [39].

It is well known that a diversified diet offers infants adequate energy, protein, and micronutrients which are essential for optimal child growth [40]. Previous evidence has demonstrated that the indicators of dietary diversity are useful proxies of micronutrient adequacy for infants [41]; thus, infants who had adequate MDD and MAD were less likely to be micronutrient-deficient. A systematic review reported consistent evidence supporting the consumption of complementary foods and beverages rich in iron (i.e., meats or iron-fortified cereal) in maintaining iron status or preventing ID during the first year of infant’s life, although there is less conclusive evidence regarding other micronutrients, such as zinc, vitamin B12, and folate [42]. In this study, we found no association between ZVF and low serum retinol. Although vitamin A-rich fruit and vegetable consumption was reported as the fourth most common food group among the eight food groups (63%), consumption was still significantly lower among infants from ethnic minorities, rural areas, and poorer families, where the prevalence of low serum retinol is more common. Additionally, while vitamin A supplementation targeting children aged 6–36 months has been implemented since 1997, the coverage among infants in poor, remote rural areas is relatively low compared to other areas [43]. These factors may have hindered progress in reducing the national prevalence of low serum retinol.

Previous cohort studies reported that infants who consumed large volumes of formula milk during early infancy had an increased risk of being overweight in later infancy and childhood [44]. This may be due to several reasons, such as the higher energy and protein content of formula milk and the unfavorable metabolic consequences of ingesting formula milk compared to breast milk (i.e., hormones, growth factors, and immunoglobulins) [45]. Thus, promoting continued breastfeeding and limiting the excess intake of formula milk during the complementary feeding stage is essential to prevent overweight children and obesity in later infancy and childhood. Our findings showed a marginal protective effect of sweet beverage consumption in reducing the odds of stunting possibly due to the associated increase in total energy intake. Other research has suggested that early exposure to sweet beverages leads to an increased preference for sweet-tasting foods and beverages, resulting in an increased risk of being overweight and childhood obesity [46]. Therefore, feeding children sweet beverages during early infancy should be avoided.

This study had many strengths. This is the first study utilizing population-based data to evaluate the impacts on DBM among Vietnamese children aged 6–23 months. Secondly, the nationally representative sample enables the findings to be generalized across Vietnam. Thirdly, the IYCF practices in this study were assessed using a wide range of global standard IYCF indicators recommended by the WHO/UNICEF guidance, which enables classification and comparison across studies. Nonetheless, this study has several limitations. Firstly, the cross-sectional study design limits our ability to establish causality, as the findings only demonstrate associations between IYCF practices and DBM and demographic or socioeconomic determinants. There may also be residual confounding factors due to unmeasured exposures. The simultaneous collection of data on IYCF practices and nutritional status prevents us from determining the temporal sequence of events, making it unclear whether inadequate IYCF practices led to DBM or vice versa, raising the possibility of reverse causation. Additionally, this design does not capture the incidence or duration of exposures. Future studies should adopt longitudinal designs to better understand the causal relationships between IYCF practices and DBM.

Second, self-reported data on IYCF practices and dietary intake may be subject to recall bias. Third, our analyses focused on only four MNDs, which may have underestimated the full prevalence of MNDs. The analysis of a broader spectrum of micronutrients including folate, iodine, vitamin B12, and vitamin D is recommended to gain a more comprehensive understanding of malnutrition. However, due to limited funding and resources, we are unable to support a bioanalytical assessment for all micronutrients. Fourth, although our study considered some of the confounding effects when defining MNDs (i.e., inflammation-adjusted serum ferritin levels for ID and adjusted cut-offs for low serum zinc based on fasting status and time of blood collection), it remains challenging to accurately analyze micronutrient biomarkers and apply cut-offs to define MNDs especially among infants and children [47]. Last but not least, this study did not capture or assess other key influencing factors of IYCF practices and DBM, such as sociocultural determinants, limiting our ability to explore these aspects thoroughly. Future research incorporating quantitative and qualitative data is warranted to address this gap and provide a more comprehensive understanding of these influences.

## 5. Conclusions

Suboptimal feeding practices and poor dietary quality of complementary foods are significant contributors to DBM among Vietnamese children. Inadequate IYCF practices were more prevalent among younger infants, ethnic minorities, poorer families, and those living in the rural and mountainous areas. Therefore, the planning and implementation of policies and programs aiming to promote IYCF knowledge and practices among mothers/caregivers and dietary diversification should consider age-specific, geographical, and socioeconomic disparities to improve IYCF and nutritional outcomes. Multi-micronutrient supplementation and fortification of complementary foods as strategies to address child malnutrition, especially in resource-deprived, remote areas, are also highly encouraged.

## Figures and Tables

**Figure 1 nutrients-16-03240-f001:**
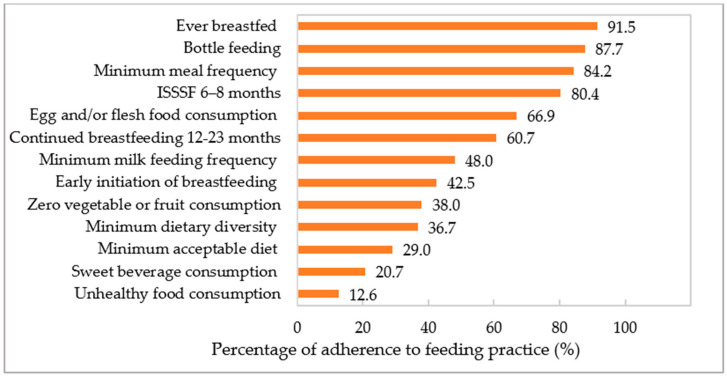
Percentage of adherence to feeding practice indicators (%) among Vietnamese children aged 6–23 months.

**Figure 2 nutrients-16-03240-f002:**
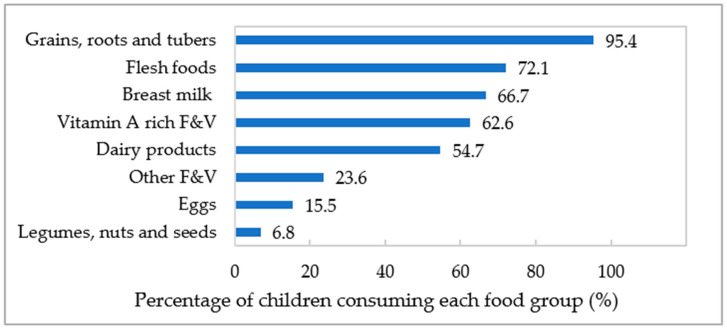
Consumption of eight food groups by Vietnamese children in the past 24 h.

**Table 1 nutrients-16-03240-t001:** Definition of feeding practice indicators based on WHO/UNICEF guidance.

Indicators	Definition
Ever breastfed (EvBF)	Proportion of children aged 6–23 months who were ever breastfed.
Early initiation of breastfeeding (EIBF)	Proportion of newborns who were put to the breast within one hour of birth.
Continued breastfeeding 12–23 months (CBF)	Proportion of children aged 6–23 months who continued breastfeeding.
Bottle feeding	Proportion of children aged 6–23 months who were bottle-fed.
Introduction of solid, semi-solid, or soft foods after 6–8 months (ISSSF)	Proportion of children aged 6–8 months who received solid, semi-solid, or soft foods.
Minimum dietary diversity (MDD)	Proportion of children aged 6–23 months who consumed at least five out of eight food groups which included (1) breast milk, (2) grains, roots, and tubers, (3) legumes, nuts, and seeds, (4) dairy products, (5) flesh foods, (6) eggs, (7) vitamin A-rich fruits and vegetables, and (8) other fruits and vegetables.
Minimum meal frequency (MMF)	Proportion of breastfed and non-breastfed children aged 6–23 months who consumed solid, semi-solid, or soft foods the minimum number of times (i.e., ≥2 times for children aged 6–8 months, ≥3 times for breastfed children aged 9–23 months, and ≥4 times for non-breastfed children).
Minimum milk feeding frequency (MMFF)	Proportion of non-breastfed children aged 6–23 months who consumed ≥ 2 milk feeds.
Minimum acceptable diet (MAD)	Proportion of children aged 6–23 months who had adequate minimum dietary diversity and minimum meal frequency.
Egg and/or flesh food consumption (EFF)	Proportion of children aged 6–23 months who consumed meats, poultry, fish, eggs, or organ meats.
Sweet beverage consumption (SWB)	Proportion of children aged 6–23 months who consumed sugar-sweetened beverages.
Unhealthy food consumption (UFC)	Proportion of children aged 6–23 months who consumed fried foods (i.e., fried dough), salty snacks (i.e., chips, crisps), and sweet snacks (i.e., sweet, ice cream, chocolate, and cakes).
Zero vegetable and fruit consumption (ZVF)	Proportion of children aged 6–23 months who did not consume vegetables or fruits.

**Table 2 nutrients-16-03240-t002:** Demographic, socioeconomic, anthropometric, and micronutrient statuses of Vietnamese children aged 6–23 months by sex (*n* = 2039).

Variables	Total (*n* = 2039)	Males (*n* = 1037)	Females (*n* = 1002)	*t*-Test/Chi Square *p*-Value
Demographic and socioeconomic indicators				
Age (months)	14 ± 5	14 ± 5	14 ± 5	t = 3.75, *p* = 0.068
Age groups				χ^2^ = 8.07, *p* = 0.117
6–11 months	730 (37.3)	347 (34.3)	383 (40.4)	
12–23 months	1309 (62.7)	690 (65.7)	619 (59.6)	
Geographical area				χ^2^ = 9.02, *p* = 0.075
Northern mountains	299 (15.0)	167 (16.8)	132 (13.1)	
Red River Delta	340 (27.6)	165 (25.7)	175 (29.4)	
North central and central coastal	365 (23.1)	174 (22.4)	191 (23.9)	
Central highlands	334 (6.1)	172 (6.4)	162 (5.9)	
Southeast	316 (14.8)	165 (15.6)	151 (13.9)	
Mekong River Delta	385 (13.4)	194 (13.0)	191 (13.8)	
Area of residence				χ^2^ = 0.02, *p* = 0.899
Urban	634 (27.5)	323 (27.7)	311 (27.4)	
Rural	1405 (72.5)	714 (72.3)	691 (72.6)	
Ethnicity				χ^2^ = 2.57, *p* = 0.134
Kinh	1619 (84.6)	814 (83.3)	805 (85.9)	
Others	420 (15.4)	223 (16.7)	197 (14.1)	
Wealth quintiles				χ^2^ = 3.10, *p* = 0.521
Poorest	308 (12.5)	169 (13.5)	139 (11.4)	
Poorer	489 (17.6)	214 (17.0)	248 (18.2)	
Middle	509 (21.9)	255 (21.1)	254 (22.8)	
Richer	490 (30.8)	250 (30.7)	240 (30.8)	
Richest	243 (17.2)	122 (17.7)	121 (16.7)	
**Anthropometric parameters**				
Height (cm)	76.9 ± 12.0	76.3 ± 5.9	77.5 ± 15.9	t = 0.22, *p* = 0.646
Weight (kg)	10.1 ± 5.4	9.7 ± 1.7	10.4 ± 7.4	t = 0.28, *p* = 0.602
Height-for-age z score (HAZ)	−0.45 ± 1.37	−0.57 ± 1.39	−0.33 ± 1.34	t = 8.25, *p* = 0.010
Stunting (HAZ < −2 SD)	213 (10.9)	136 (14.0)	77 (7.7)	χ^2^ = 17.67, *p* = 0.001
Weight-for-age z score (WAZ)	−0.35 ± 1.15	−0.40 ± 1.20	−0.30 ± 1.09	t = 2.29, *p* = 0.146
Underweight (WAZ < −2 SD)	117 (5.6)	76 (7.2)	41 (3.9)	χ^2^ = 8.75, *p* = 0.003
Weight-for-height z score (WHZ)	−0.16 ± 1.06	−0.14 ± 1.10	−0.19 ± 1.01	t = 1.61, *p* = 0.219
Normal-weight (−2 SD ≤WHZ ≤ + 2 SD)	1617 (93.6)	816 (93.5)	801 (93.7)	χ^2^ = 0.56, *p* = 0.836
Wasting (WHZ < −2 SD)	64 (3.4)	37 (3.6)	27 (3.2)	
Overweight (+2 SD < WHZ ≤ +3 SD)	42 (2.5)	21 (2.3)	21 (2.6)	
Obesity (WHZ > +3 SD)	10 (0.6)	7 (0.7)	3 (0.5)	
**Biomarkers of micronutrients**				
Hemoglobin (g/dL)	11.4 ± 1.1	11.3 ± 1.2	11.4 ± 1.1	t = 1.24, *p* = 0.279
Serum ferritin (µg/L) ^a^	19.3 ± 0.6	17.2 ± 0.7	21.8 ± 1.0	t = 21.32, *p* < 0.001
Serum transferrin receptor (mg/L)	8.1 ± 3.8	8.6 ± 4.1	7.5 ± 3.2	t = 30.45, *p* < 0.001
Body iron store	1.9 ± 4.4	1.3 ± 4.4	2.6 ± 4.3	t = 20.42, *p* < 0.001
Serum zinc (µmol/L)	9.7 ± 6.1	9.7 ± 6.1	9.7 ± 6.1	t = 0.001, *p* = 0.979
Serum retinol (µmol/L)	1.0 ± 0.3	1.0 ± 0.3	1.0 ± 0.3	t = 1.57, *p* = 0.226
Retinol-binding protein (µmol/L)	1.0 ± 0.4	1.0 ± 0.4	1.0 ± 0.4	t = 1.83, *p* = 0.192
C-reactive protein (mg/L)	1.9 ± 5.7	2.1 ± 6.3	1.6 ± 4.9	t = 2.17, *p* = 0.158
Alpha-1 acid glycoprotein (mg/L)	0.67 ± 0.3	0.69 ± 0.4	0.65 ± 0.3	t = 5.19, *p* = 0.035
**Micronutrient deficiencies (MNDs)**				
Anemia	583 (31.2)	310 (32.4)	273 (29.9)	χ^2^ = 1.33, *p* = 0.202
Iron deficiency	626 (34.6)	379 (40.0)	247 (28.8)	χ^2^ = 23.93, *p* = 0.009
Iron deficiency anemia	347 (17.8)	212 (21.0)	135 (14.3)	χ^2^ = 13.80, *p* = 0.001
Low serum zinc	1066 (56.7)	549 (56.1)	517 (57.5)	χ^2^ = 0.36, *p* = 0.499
Low serum retinol	254 (14.3)	24 (14.3)	123 (14.3)	χ^2^ = 0.0003, *p* = 0.989
Total number of MNDs *				
0–1	981 (48.9)	467 (46.0)	514 (52.0)	χ^2^ = 7.29, *p* = 0.018
2–4	1058 (51.1)	570 (54.0)	488 (48.0)	
Inflammation	335 (18.5)	182 (19.4)	153 (17.4)	χ^2^ = 1.17, *p* = 0.331

Continuous and categorical variables are reported as mean ± standard deviation and number (percentage), respectively. Differences in continuous and categorical variables between males and females were assessed using the *t*-test and Chi-square test of association, respectively. ^a^ Geometric mean and standard error are reported for non-normally distributed variables. * Total number of MNDs included anemia, iron deficiency, low serum zinc, and low serum retinol. *p* < 0.05 was considered as significantly different. HAZ, height-for-age z score; WAZ, weight-for-age z score; WHZ, weight-for-height z score.

**Table 3 nutrients-16-03240-t003:** Unadjusted binomial logistic regressions between feeding practices and demographic and socioeconomic determinants among Vietnamese children.

Variables		Early Initiation of Breastfeeding	Continued Breastfeeding (12–23 Months)	Minimum Dietary Diversity	Minimum Meal Frequency	Minimum Milk Feeding Frequency	Minimum Acceptable Diet	Egg and/or Flesh Foods Consumption	Sweet Beverage Consumption	Unhealthy Food Consumption	Zero Vegetable or Fruit Consumption
		OR (95% CI)	OR (95% CI)	OR (95% CI)	OR (95% CI)	OR (95% CI)	OR (95% CI)	OR (95% CI)	OR (95% CI)	OR (95% CI)	OR (95% CI)
Age (Ref: 6–11 months)	12–23 months			1.91 (1.49, 2.46) ***		4.26 (3.00, 6.06) ***	2.15 (1.61, 2.86) ***	1.69 (1.26, 2.25) ***	2.52 (1.60, 3.95) ***	3.37 (2.53, 4.49) ***	
Area of residence (Ref: urban)	Rural		0.98 (0.68, 1.40)	0.65 (0.49, 0.86) **			0.77 (0.60, 0.98) *	1.02 (0.76, 1.38)			
	Mountainous		0.40 (0.22, 0.71) **	0.44 (0.26, 0.74) **			0.48 (0.23, 1.00)	0.52 (0.32, 0.85) *			
Ethnicity (Ref: Kinh major)	Minorities		0.39 (0.21, 0.71) **	0.65 (0.54, 0.79) ***		0.56 (0.34, 0.91) *	0.36 (0.20, 0.64) **	0.41 (0.29, 0.57) ***			1.86 (1.32, 2.63) ***
Wealth quintiles (Ref: poorest)	Poorer		1.84 (0.92, 3.68)	1.19 (0.66, 2.15)	1.50 (0.94, 2.40)			1.55 (0.88, 2.70)			0.85 (0.50, 1.43)
	Middle		2.19 (0.95, 5.03)	1.65 (0.94, 2.90)	1.67 (0.98, 2.86)			2.76 (1.72, 4.41) ***			0.56 (0.35, 0.89) *
	Richer		2.79 (1.22, 6.35) *	1.80 (0.97, 3.32)	2.43 (1.29, 4.57) **			2.82 (1.51, 5.28) **			0.49 (0.30, 0.78) **
	Richest		2.32 (1.04, 5.19) *	2.69 (1.28, 5.67) *	2.23 (1.41, 3.54) **			2.89 (1.72, 4.83) ***			0.42 (0.25, 0.70) **
Geographical area (Ref: Northern mountains)	Red River Delta	1.93 (0.30, 12.34)	2.54 (1.14, 5.65) *	1.81 (1.06, 3.10) *				1.64 (0.79, 3.40)	0.67 (0.29, 1.53)		
	North central and central coastal	5.20 (1.22, 22.15) *	1.81 (0.87, 3.75)	1.04 (0.49, 2.22)				2.19 (1.02, 3.96) *	0.45 (0.20, 0.99) *		
	Central highlands	4.08 (0.70, 23.77)	1.33 (0.34, 5.24)	0.89 (0.26, 3.05)				0.77 (0.32, 1.82)	0.41 (0.13, 1.38)		
	Southeast	2.17 (0.61, 7.77)	2.63 (1.42, 4.89) ***	2.39 (1.36, 4.21) **				1.69 (0.77, 3.72)	0.80 (0.33, 1.99)		
	Mekong River Delta	5.88 (1.61, 21.40) **	2.63 (1.42, 4.89) **	1.02 (0.54, 1.94)				0.98 (0.37, 2.60)	0.91 (0.37, 2.23)		

Bivariate logistic regressions were performed and reported as ORs with a respective 95% CI. * *p* < 0.05; ** *p* < 0.01; *** *p* < 0.001. CI, confidence intervals; OR, odds ratio; Ref: reference group. No significant association was found with sex and introduction of solid, semi-solid, or soft foods at 6–8 months; thus, these findings were not reported.

**Table 4 nutrients-16-03240-t004:** Adjusted binomial logistic regressions between feeding practices and malnutrition among Vietnamese children.

Variables	Stunting (HAZ < −2 SD)	Underweight (WAZ < −2 SD)	Wasting (WHZ < −2 SD)	Overweight (WHZ > +2 SD)
	AOR (95% CI)	AOR (95% CI)	AOR (95% CI)	AOR (95% CI)
Continued breastfeeding (12–23 months)		0.50 (0.27, 0.92) *		
Introduction of solid, semi-solid, or soft foods at 6–8 months			0.13 (0.04, 0.40) ***	
Adequate minimum dietary diversity	0.61 (0.41, 0.92) *			
Adequate minimum milk feeding frequency				3.33 (1.01, 11.09) *
Egg and/or flesh foods consumption	0.68 (0.48, 0.97) *			
Sweet beverage consumption	0.64 (0.42, 0.99) *			
Zero vegetable or fruit consumption	1.55 (1.18, 2.04) **			

Multivariate logistic regression analyses were performed and reported as AOR (95% CI). The “No” category for each feeding practice was used as the reference group. Models were adjusted by age, sex, area of residence, wealth quintiles, and inflammation. * *p* < 0.05; ** *p* < 0.01; *** *p* < 0.001. AOR, adjusted odds ratio; CI, confidence interval; HAZ, height-for-age z score; WAZ, weight-for-age z score; WHZ, weight-for-height z score.

**Table 5 nutrients-16-03240-t005:** Adjusted binomial logistic regressions between feeding practices and different forms of micronutrient deficiencies among Vietnamese children.

Variables	Anemia	Iron Deficiency	Iron Deficiency Anemia	Low Serum Zinc	Low Serum Retinol
	AOR (95% CI) ^b^	AOR (95% CI) ^a^	AOR (95% CI) ^a^	AOR (95% CI) ^b^	AOR (95% CI) ^b^
Continued breastfeeding (12–23 months)		0.75 (0.58, 0.98) *		0.70 (0.52, 0.96) *	
Introduction of solid, semi-solid, or soft foods at 6–8 months					0.37 (0.16, 0.89) *
Adequate minimum dietary diversity		0.69 (0.54, 0.88) **	0.63 (0.46, 0.88) **		
Adequate minimum milk feeding frequency		0.56 (0.38, 0.82) **			
Adequate minimum acceptable diet	0.72 (0.57, 0.91) **	0.66 (0.52, 0.84) **	0.56 (0.42, 0.75) ***		0.63 (0.41, 0.99) *

Multivariate logistic regression analyses were performed and reported as AOR (95% CI). The “No” category for each feeding practice was used as the reference group. ^a^ adjusted by age, sex, area of residence, and wealth quintiles; ^b^ model ^a^ and inflammation. * *p* < 0.05; ** *p* < 0.01; *** *p* < 0.001. AOR, adjusted odds ratio; CI, confidence interval.

## Data Availability

The datasets presented in this article are not publicly available. This dataset is the property of the National Institute of Nutrition (NIN), Vietnam, and is available for research purposes on reasonable request. Requests to access the dataset should be directed to ninvietnam@viendinhduong.vn.

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
