# Peer review of "The Role of Complementary Feeding Practices in Addressing the Double Burden of Malnutrition among Children Aged 6–23 Months: Insight from the Vietnamese General Nutrition Survey 2020"

_nutrients, 2024, doi:10.3390/nu16193240_

Round 1
Reviewer 1 Report
Comments and Suggestions for Authors
There is value in this paper and its analysis, but I do have some reflections mainly related to the methods reporting approach taken.
It is not sufficiently clear to this reviewers on ‘how’ IYCF indicators and outcome variables were ‘selected’. The same holds true for the blood analysis dataset, how have these parameters been selected.
Were all 17 standard indicators assessed ? In the full paper, it is suggested that 13 ‘key’ indicators were examined, but it is not clear why the other 4 were omitted, and what the rationale is to do so.
One the biochemical assessment, you refer to another paper, but this does not make clear why and how these markers have been selected.
On 2.6, how have quintiles been generated, where these based on external reference data, or were there driven by the data as collected ?
Specific comments
The (double) burden of malnutrition should be better described in the abstract. In the first sentence of the paper, a triple burden is suggested. Please harmonize, and be explicit.
The reference 7 is not good described, as this looks like a part (?chapter) of the document.
On the 2.5 section, perhaps a table provides more overview ?
I miss a statement on changes over time (versus the 2011 project) in the conclusions section.
Author Response
Reviewer 1
There is value in this paper and its analysis, but I do have some reflections mainly related to the methods reporting approach taken.
Reviewer's comments |
Authors’ responses |
Comment 1: It is not sufficiently clear to this reviewer on ‘how’ IYCF indicators and outcome variables were ‘selected’. The same holds true for the blood analysis dataset, how have these parameters been selected. |
Thank you for reviewer’s valuable comments. We have addressed the reviewer’s concerns and refined our methodology section accordingly. The justifications for the selection of the 13 IYCF indicators (please refer to Comment 2); and the micronutrient deficiencies/biomarkers (anemia, iron deficiency, iron deficiency anemia, low serum retinol and low serum zinc) (please refer to Comment 3) have now described and discussed in the section below. |
Comment 2: Were all 17 standard indicators assessed ? In the full paper, it is suggested that 13 ‘key’ indicators were examined, but it is not clear why the other 4 were omitted, and what the rationale is to do so. |
Of those 17 IYCF indicators recommended by WHO/UNICEF, we selected 13 indicators that assess IYCF practices for infants aged 6-23 months. The excluded four indicators (i.e. exclusive breastfeeding under six months, mixed milk feeding under six months, exclusively breastfed for the first two days after birth, and infant feeding area graph) pertain to breastfeeding practices for infants aged 0-5 months and are not relevant to our study population, therefore were omitted. This explanation has now been included in the Methodology Section (Section 2.4, Lines 164-169). |
Comment 3: One the biochemical assessment, you refer to another paper, but this does not make clear why and how these markers have been selected. |
The blood micronutrients investigated in this study were selected based on three main criteria: 1) micronutrient biomarkers that were measured and available in the GNS 2020, 2) high prevalence and severity of deficiencies for these micronutrients in Vietnam, and 3) alignment with the priorities of Vietnamese government based on the nutritional indicators and targets outlined in the National Nutrition Strategy 2025 and 2030. As a result, four micronutrient deficiencies including anemia, iron deficiency, vitamin A deficiency (now referred as “low serum retinol”), and zinc deficiency (now referred as “low serum zinc”) were selected. The rationale for selecting these four micronutrients have now been described in the Methodology Section (Section 2.3, Lines 127-135). |
Comment 4: On 2.6, how have quintiles been generated, where these based on external reference data, or were there driven by the data as collected ? |
Thank you for your comment. The household wealth index score was calculated using data on household’s ownership of selected assets, materials used for housing construction, types of water access, and sanitation facilities collected in our survey, using principal component analysis (PCA) following the Demographic and Health Survey (DHS) guidelines. Each household was then assigned a wealth index score based on the PCA factor loadings and was further grouped into quintiles for analysis: poorest (Q1), poorer (Q2), middle (Q3), richer (Q4), and richest (Q5) quintiles. This methodology is now detailed in the Methodology Section (Section 2.5, Lines 176-181). |
Comment 5: The (double) burden of malnutrition should be better described in the abstract. In the first sentence of the paper, a triple burden is suggested. Please harmonize, and be explicit. |
Thank you for the suggestion. The double burden of malnutrition (DBM) has now been described in the abstract (Line 30-32). Additionally, we have ensured that DBM is used consistently throughout the manuscript, and reference to the term ‘TBM’ has been removed. |
Comment 6: The reference 7 is not good described, as this looks like a part (?chapter) of the document. |
The reference (revised as reference 8) has now been revised as per the WHO/UNICEF recommendation and journal guideline. |
Comment 7: On the 2.5 section, perhaps a table provides more overview ? |
Thank you for the suggestion. The IYCF indicators and their definition were initially reported in Supplementary Table 1. This table has now been incorporated into the main manuscript as Table 1. |
Comment 8: I miss a statement on changes over time (versus the 2011 project) in the conclusions section. |
Thank you for the reviewer’s comment. We compared our findings with the 2011 A&T Project for context, these comparisons should be interpreted cautiously as they do not reflect longitudinal trends. Since our study is cross-sectional, we steer away from drawing conclusion about trends or changes in prevalence over time. As such, we did not include any statement regarding changes over time in the conclusion. The comparison is addressed in the discussion section, where we note methodological differences and their implications. We hope this adequately addresses the reviewer’s concern. |
Reviewer 2 Report
Comments and Suggestions for Authors
The reviewed article is an interesting study on complementary feeding practices and their impact on the dual burden of malnutrition among children aged 6 to 23 months in Vietnam. The authors provide detailed data on dietary diversity and dietary quality, and analyse the demographic and socio-economic determinants of these practices. The findings are valuable and provide important information that can help shape nutrition policy not only in Vietnam, but also in other countries with similar contexts.
However, the article is not without limitations. Firstly, it is a cross-sectional study, which limits the ability to draw causal conclusions. While the authors themselves acknowledge this, the article lacks a more detailed discussion of the potential flaws and limitations of such an approach. Furthermore, the analysis was limited to the four main micronutrient deficiencies, which may lead to an underestimation of the full extent of the malnutrition problem. It would have been worth considering including a broader spectrum of micronutrients or at least discussing why these specific indicators were chosen.
Another limitation is the lack of a deeper analysis of the cultural and social determinants that influence dietary practices, especially in the context of differences between ethnic groups. Although the article mentions some traditional beliefs, a more comprehensive analysis of these factors and their impact on dietary diversity and dietary quality is missing.
I recommend that the authors supplement it with a more detailed discussion of the limitations of the study, including the impact of the methodological approach used on the results. It is also necessary to consider the inclusion of a broader analysis of micronutrients or, at least, a more detailed justification for the selection of only four indicators. Finally, it would be worthwhile to deepen the analysis of the socio-cultural determinants of dietary practices in order to better understand the context and variation in outcomes between different demographic groups. Amendments of this kind could significantly strengthen the article and make it more complete and valuable for readers.
Author Response
Reviewer 2:
The reviewed article is an interesting study on complementary feeding practices and their impact on the dual burden of malnutrition among children aged 6 to 23 months in Vietnam. The authors provide detailed data on dietary diversity and dietary quality, and analyse the demographic and socio-economic determinants of these practices. The findings are valuable and provide important information that can help shape nutrition policy not only in Vietnam, but also in other countries with similar contexts.
Reviewer's comments |
Authors’ responses |
Comment 1: I recommend that the authors supplement it with a more detailed discussion of the limitations of the study, including the impact of the methodological approach used on the results. Firstly, it is a cross-sectional study, which limits the ability to draw causal conclusions. While the authors themselves acknowledge this, the article lacks a more detailed discussion of the potential flaws and limitations of such an approach. |
Thank you for your valuable suggestions. We have now included a detailed discussion of the limitations of the study, in terms of the study design and approach (Discussion Section, Lines 423-431), as below: “…First, the cross-sectional study design limits our ability to establish causality, as the findings only demonstrate associations between IYCF practices and DBM, and demo-graphic or socioeconomic determinants. There may also be residual confounding due to unmeasured exposures. The simultaneous collection of data on IYCF practices and nutritional status prevents us from determining the temporal sequence of events, making it unclear whether inadequate IYCF practices led to DBM or vice versa, raising the possibility of reverse causation. Additionally, this design does not capture the incidence or duration of exposures. Future studies should adopt longitudinal designs to better understand the causal relationships between IYCF practices and DBM.” |
Comment 2: It is also necessary to consider the inclusion of a broader analysis of micronutrients or, at least, a more detailed justification for the selection of only four indicators. Furthermore, the analysis was limited to the four main micronutrient deficiencies, which may lead to an underestimation of the full extent of the malnutrition problem. It would have been worth considering including a broader spectrum of micronutrients or at least discussing why these specific indicators were chosen. |
We agree that limited to the only four micronutrient deficiencies may lead to an underestimation of the full extent of malnutrition. We have now included a detailed justification for the selection of the four indicators in the Methodology Section (Section 2.3, Lines 127-135) as below: “The blood micronutrients investigated in this study were selected based on three main criteria: 1) micronutrient biomarkers that were measured and available in the GNS 2020 dataset, 2) high prevalence and severity of deficiencies for these micronutrients in Vietnam, and 3) alignment with the priorities of Vietnamese government based on the nutritional indicators and targets outlined in the National Nutrition Strategy 2025 and 2030. As a result, four micronutrient deficiencies including anemia, iron deficiency, vitamin A deficiency (now referred as “low serum retinol”), and zinc deficiency (now referred as “low serum zinc”) were selected. These were reviewed and agreed by the advisory panel from the NIN, Vietnam.” We have also included the statement below in the Limitations Section (Lines 434-437).: “Third, our analyses focused only on four MNDs, which may have underestimated the full prevalence of MNDs. Analysis of a broader spectrum of micronutrients including folate, iodine, vitamin B12 and vitamin D is recommended to gain a more comprehensive understanding of mal-nutrition. However, due to limited funding and resources, we are unable to support bioanalytical assessment for all micronutrients.”
|
Comment 3: Finally, it would be worthwhile to deepen the analysis of the socio-cultural determinants of dietary practices in order to better understand the context and variation in outcomes between different demographic groups. Another limitation is the lack of a deeper analysis of the cultural and social determinants that influence dietary practices, especially in the context of differences between ethnic groups. Although the article mentions some traditional beliefs, a more comprehensive analysis of these factors and their impact on dietary diversity and dietary quality is missing. |
Thank you for your insightful suggestion. We agree with the reviewer on the importance of socio-cultural determinants in shaping mothers’ feeding practices and their subsequent impact on child nutritional outcomes, particularly across different demographic or ethnic groups. Our data did not capture detailed information on these practices and beliefs, limiting our ability to explore these aspects more thoroughly. We have now included this point in the Limitations Section (Line 442-446). Additionally, we have provided a more in-depth discussion on the possible effects of socio-cultural determinants on mothers’ feeding practices, and emphasised the importance of incorporating these considerations in future research and interventions aimed at improving IYCF practices and child nutritional outcomes, particularly in resources-constrained and culturally diverse settings in Vietnam (Discussion Section, Lines 358-373). We believe these additions strengthen the manuscript and provide a more comprehensive understanding of the context in which IYCF practices occur.
|
Round 2
Reviewer 1 Report
Comments and Suggestions for Authors
the authors have addressed my comments and concerns